# Distributed Deep Features Extraction Model for Air Quality Forecasting

**Axel Gedeon Mengara Mengara** [1] , **Younghak Kim** [1] , **Younghwan Yoo** [1,]*  **and Jaehun Ahn** [2]

1    School of Computer Science and Engineering, Pusan National University, Busan 46241, Korea;
mengara@pusan.ac.kr (A.G.M.M.); qwdxzokm1@pusan.ac.kr (Y.K.)
2    School of Urban, Architecture and Civil Engineering, Pusan National University, Busan 46241, Korea;
jahn@pusan.ac.kr
*    Correspondence: ymomo@pusan.ac.kr

**Abstract:** Several studies in environmental engineering emphasize the importance of air quality forecasting for sustainable development around the world. In this paper, we studied a new approach for air quality forecasting in Busan metropolitan city. We proposed a convolutional Bi-Directional Long-Short Term Memory (Bi-LSTM) autoencoder model trained using a distributed architecture to predict the concentration of the air quality particles ($PM_{2.5}$ and $PM_{10}$). The proposed deep learning model can automatically learn the intrinsic correlation among the pollutants in different location. Also, the meteorological and the pollution gas information at each location are fully utilized, which is beneficial for the performance of the model. We used multiple one-dimension convolutional neural network (CNN) layers to extract the local spatial features and a stacked Bi-LSTM layer to learn the spatiotemporal correlation of air quality particles. In addition, we used a stacked deep autoencoder to encode the essential transformation patterns of the pollution gas and the meteorological data, since they are very important for providing useful information that can significantly improve the prediction of the air quality particles. Finally, in order to reduce the training time and the resource consumption, we used a distributed deep leaning approach called *data parallelism*, which has never been used to tackle the problem of air quality forecasting. We evaluated our approach with extensive experiments based on the data collected in Busan metropolitan city. The results reveal the superiority of our framework over ten baseline models and display how the distributed deep learning model can significantly improve the training time and even the prediction accuracy.

**Keywords:** air quality prediction; deep learning; $PM_{2.5}$; $PM_{10}$; Busan city; distributed model

## 1. Introduction

Nowadays, people are very concerned about air pollution with the development of industries. The concentration of various kinds of pollution gas and solid particle, such as $PM_{2.5}$, $PM_{10}$, $NO_2$, $SO_2$, $CO$, and $O_3$ impacts human health and sustainable development around the world. In South Korea, research on air pollution is very important and has constantly been viewed as a key topic in environmental protection [1]. As depicted in Figure 1, air pollution has several significant factors. We can classify these factors into two specific categories, namely primary factors and secondary factors. The primary factors are essentially based on air pollutants such as solid particles, coal burning, traffic volumes, and manufacturing emission. Each of these sources has a different spatial distribution and temporal pattern. On the other hand, the secondary factors are mainly composed of meteorological information, topography, and time. There is an increasing demand for predicting future air quality, because people can take more precautions in order to not get sick if they know the air quality in advance. Air quality forecasting is also highly significant to any government's emergency management, since

it can provide time for the government to implement appropriate emergency measures to mitigate atmospheric pollution, such as limiting the production and emissions of heavily polluting enterprises and restricting motor vehicles. However, air quality prediction is a complex task and improving the accuracy of predictions and reducing the training time is an urgent and challenging problem in the field of air pollution prevention.

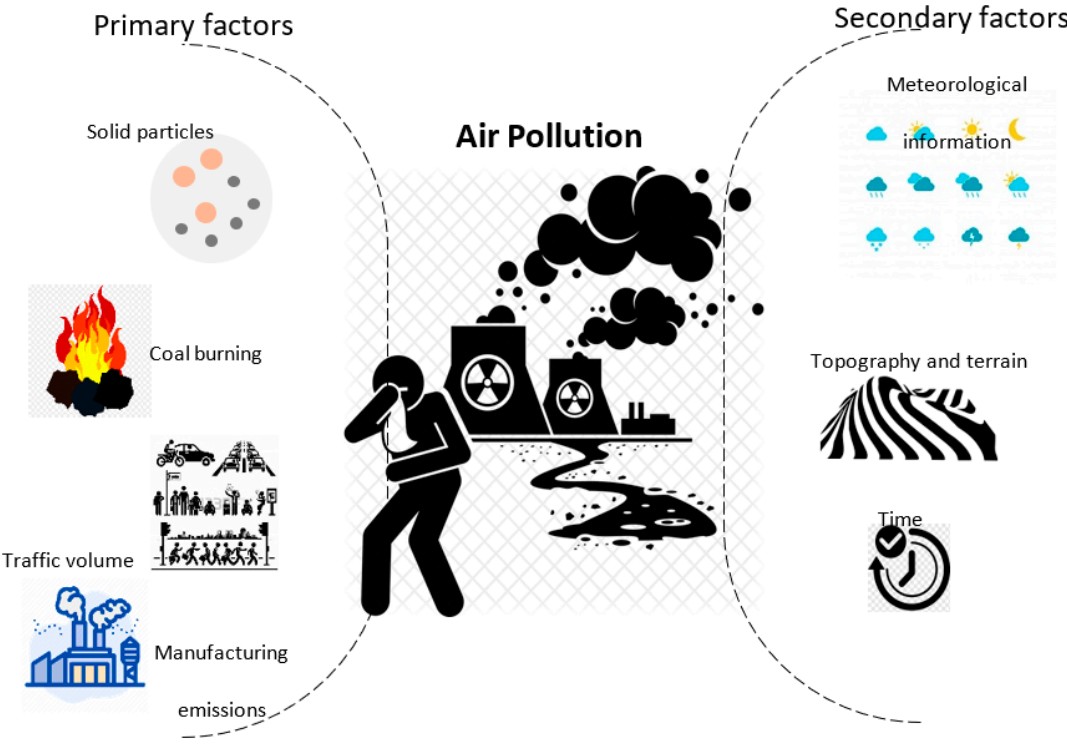

**Figure 1.** Air pollution factors.

For the past ten years, several researchers have made effort on the air quality forecasting topic. In these studies, the authors mainly focused on two types of modeling for air quality prediction. The first type is knowledge-based models and the second type is data-driven models. The knowledge-based models mainly focus on chemical and physical assumptions to represent the transportation and transformation of air pollution particles. Many knowledge-based models have been proposed in the literature. However, the successful application of knowledge-based models requires a solid background in atmospheric and environmental science. Furthermore, when the model is applied in different situations, the chemical and transportation rules may change, which generally lead the model to inaccurate results. To solve this problem, some researchers in the literature implemented statistical prediction methods such as autoregressive integrated moving average (ARIMA) [2], hidden semi-Markov models (HSMMs) [3], and least absolute shrinkage and selection operator (LASSO) model [4]. However, the statistical prediction method, which implements a mathematical logic and regression analysis, has two significant shortcomings: (1) low accuracy; (2) urge time and energy consumption, mainly caused by the analysis of long-term historical monitoring data. To summarize, the statistical prediction methods [5] are effective for air quality forecasting. However, the diversity factors of air pollution make it difficult to achieve a good prediction accuracy.

Recently, with the explosion of the era of big data and artificial intelligence, the data-driven approach for air pollution modeling has been considered and applied in several forecasting systems. Air quality prediction methods based on machine learning algorithms have overcome some of the shortcomings of the older statistical prediction methods and numerical predictions mentioned above and have become the mainstream of air quality prediction research. So far, air quality prediction

methods based on machine learning have achieved some good results. For instance, Wang et al. [6] proposed an online support vector machine (SVM) model to predict air pollutant concentration in the Hong Kong downtown area. They performed a comparative experiment between conventional SVM and online SVM and demonstrated the effectiveness of their model. In study [7], the authors made a case study for air pollution prediction in Murcia city. The authors focused on the prediction of the ozone ($O_3$) level. They used several shallow machine learning models. Among these models the Random Forest performed the best. Recently, study [8] proposed an air pollution forecasting approach by using four advanced regression techniques (Decision Tree Regression, Random Forest Regression, Gradient Boosting Regression, Artificial Neural Network (ANN) Multi-Layer Perceptron Regression) and presented a comparative study to determine the best model for accurately predicting air quality with reference to data size and processing time. The experiment results showed that Random Forest Regression was the best model, performing well for pollution prediction for data sets of varying size and location and with different characteristics.

Deep learning (DL) has been extremely applied in big data analysis to solve several problems related to object recognition [9], image classification [10], speech recognition [10], time series forecasting [10], and so on. Furthermore, the advent of deep learning technologies has remarkably enhanced the accuracy and efficiency of air quality prediction. Deep learning is currently the most popular data-driven method [9], which can extract and learn the inherent features of various air quality data automatically. A wide range of papers applying deep learning for air pollution prediction in literature have achieved good results. Among these papers, Zhao et al. [11] proposed a deep learning model called as long short-term memory—fully connected (LSTM-FC) neural network, to predict the concentration of ***PM*₂.₅** among specific monitoring stations over 48 hours. The authors used historical air quality data, meteorological data, and weather forecast data as input for their prediction model. Finally, they evaluated the proposed approach with a dataset containing records of 36 air quality monitoring stations and made a comparison with an ANN model and an LSTM model on the same dataset. Qi et al. [12] proposed a hybrid model based on deep learning methods that embeds graph convolutional networks and long short-term memory networks (GC-LSTM) to model and predict the spatiotemporal variation of ***PM*₂.₅** concentrations. The authors constructed historical observations as spatiotemporal graph series, and historical air quality variables, meteorological factors, spatial terms, and temporal attributes were defined as graph signals. For evaluation purposes, the authors compared their model with some state-of-the-art approaches in different time intervals and based on the results of the proposed model, achieved the best performance for predictions. Wen et al. [13] proposed a convolutional long short-term memory neural network model to predict the concentration level of ***PM*₂.₅**. Spatiotemporal features were extracted through the combination of the convolutional neural network (CNN) and LSTM network. The meteorological data and aerosol data were also integrated, in order to improve the performance of the model. Similar to the previous study, Huang et al. [14] developed a model that integrates CNN and LSTM for ***PM*₂.₅** prediction. They evaluated their model by using four measurement indexes (Mean Absolute Error, Root Mean Square Error, Pearson correlation coefficient, and Index of Agreement) in the experiments. Bai et al. [15] proposed a stacked autoencoder model combining seasonal analysis and deep feature learning to predict the hourly concentration of ***PM*₂.₅**. They evaluated their model using a dataset collected from three environmental monitoring stations in Beijing. The results demonstrated the effectiveness of the proposed approach. Wang et al. [16] used a hybrid deep learning model based on CNN and seq2seq. The CNN layer was used to extract the spatial correlation among different stations and the seq2seq to capture the temporal relationship for final prediction.

As we can see, a lot of hybrid models have been used recently for air quality forecasting task, and some of them are performing well. However, these models suffer from two significant problems:

- The first problem is the very slow training speed, since the huge amount of data coming from the different air quality monitoring stations is trained by using a centralized deep learning architecture. In some worst cases these models need retraining because of their degradation caused by the

variation of data distribution over time. So, the problem of the time and resource-consuming during the training step is definitely a big challenge in air quality forecasting;

- The second shortcoming of these approaches is that they do not consider the fact that there is usually some noise in air quality data and meteorological data, which affects, to a certain extent, the accuracy and the performance of their predictions, since they are not able to extract suitable features and information from pollution gas and meteorological data.

Considering these challenges, in this research we propose a deep learning model based on a convolutional Bi-LSTM autoencoder framework for air quality forecasting. The proposed model is trained by using a distributed architecture called *data parallelism*. The main contributions of this paper are as follows:

- Study of current state-of-the-art machine learning and deep learning approaches for air quality forecasting;
- Design and implementation of a distributed deep learning approach based on two-stage feature extraction for air quality prediction. In the first stage, a stacked autoencoder extracts useful features and information from pollution gas data and meteorological data. In the second stage, considering the properties of multivariate time series air quality particles data, we use a one-dimension convolutional layer (1D-CNN) to extract the local pattern features and the deep spatial correlation features from air quality particles data. The CNN model is widely used in object recognition and image processing area, but due to its one-dimensional characteristics it can also be applied to time series forecasting tasks. Finally, the extracted features are interpreted by a Bi-LSTM layer throughout time steps to make the final prediction;
- Evaluation of the proposed approach based on two specific phases. In the first phase we train our deep learning framework within a centralized architecture with a single training server and we compare it against ten state-of-the-art models. In the second phase we use a distributed deep learning architecture called *data parallelism* to train the proposed framework on several training workers to optimize its accuracy and its training time.

The remaining parts of this paper are organized as follows: Section 2 presents the data collection and the feature correlation. In Section 3 we introduce our deep learning framework. The experiments are described in Section 4. Finally, the conclusion and future work are discussed in Section 5.

## 2. Data Collection and Features Correlation

### 2.1. Data Collection

The proposed deep learning approach was tested by using two different data sources: (1) air quality monitoring data from Air Korea [17] and (2) meteorological data from the Korean Meteorological Agency (KMA) [18]. Both data sources were collected in Busan metropolitan city. The air quality dataset included 20 monitoring stations, as depicted in Figure 2. The dataset period was from January 2019 to December 2019. It contained nine specific features, which were: $PM_{2.5}$, $PM_{10}$, $NO_2$ (Nitrogen dioxide), $CO$ (Carbon monoxide), $O_3$ (Ozone), $SO_2$ (Sulfur dioxide) for air quality related features, and the temperature, the humidity, and the wind speed for meteorological features.

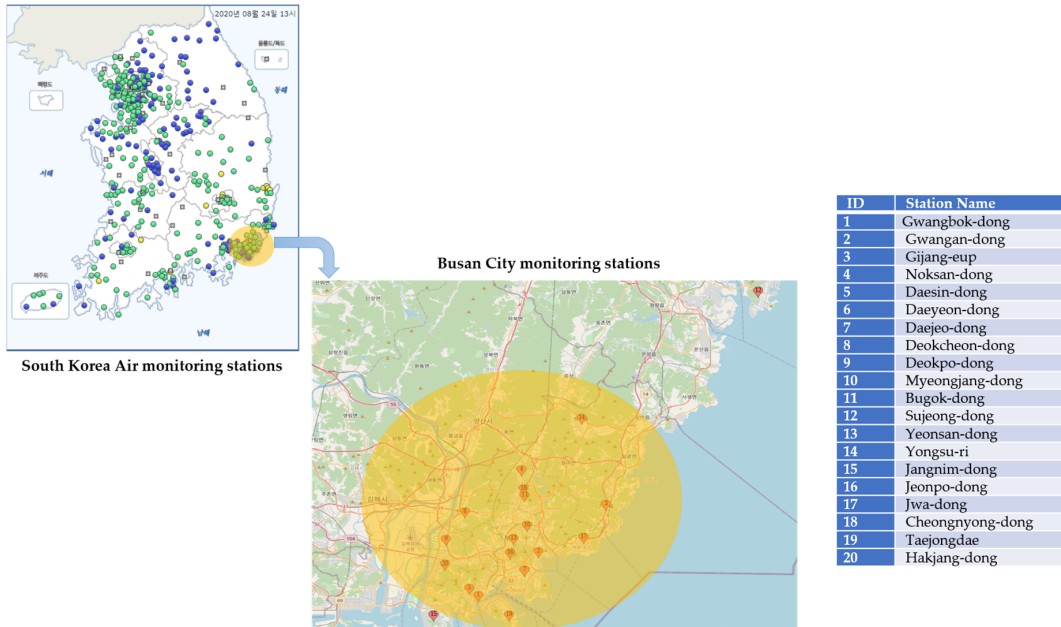

**Figure 2.** Air quality monitoring station present in Busan city.

## 2.2. Statistical Analysis

Figure 3 presents the overall characteristics of the air quality data. It shows the mean values of the **PM<sub>2.5</sub>** and the **PM<sub>10</sub>** particles for all the monitoring stations except stations 2, 10, 16, 17, and 18, because those stations had several months of missing values, and were excluded from this study.

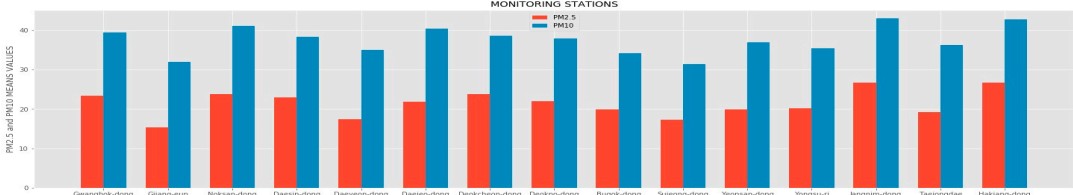

**Figure 3.** $PM_{2.5}$ and $PM_{10}$ concentrations.

Among them, Jangnim-dong station and Hakjang-dong station recorded, respectively, 26.63 ug/m$^3$ and 26.6 ug/m$^3$ for **PM<sub>2.5</sub>** and 46.21 ug/m$^3$ and 46.41 ug/m$^3$ for **PM<sub>10</sub>**, which were reported as the highest mean values among all the stations. This result also shows that Busan city did not suffer from very bad air quality during 2019, since according to the Korean air quality index (CAI) [17], if the **PM<sub>2.5</sub>** concentration is between 16 and 35 and the **PM<sub>10</sub>** concentration between 31 and 80, this means that the CAI is normal.

As mentioned above, we did not consider some stations due to their huge amount of missing values. Then, the remaining 15 stations also had some missing values due to sensor errors. Without changing the design of the proposed framework, we used the average value of the data in the same period to replace the missing values based on the following equations:

$$x_t^{\beta} \Leftarrow m_t^{\beta} x_t^{\beta} + \left(1 - m_t^{\beta}\right)\tilde{x}^{\beta}, \tag{1}$$

$$m_t^{\beta} = \begin{cases} 1, x_t^{\beta}, & True \\ 0, x_t^{\beta}, & False \end{cases}, \tag{2}$$

$$\frac{\sum_{\alpha=1}^{T} m_{\alpha}^{\beta}, x_{\alpha}^{\beta}, H_{hour}(S_t, S_\alpha)}{\sum_{\alpha=1}^{T} m_{\alpha}^{\beta}, H_{hour}(S_t, S_\alpha)},$$ (3)

$$H_{hour}(S_t, S_\alpha) = \begin{cases} 1, & S_t = S_\alpha \\ 0, & S_t \neq S_\alpha \end{cases}.$$ (4)

In the above equations, $\alpha$ represents the time step where the $\beta$-dimensional component has been observed, $\widetilde{x}^{\beta}$ is the mean value of the $\beta$-dimensional component of the current time in the same month, $x_t^{\beta}$ represents the current true observation of the component, and $S_t$ is the time corresponding to the *t*-th time step. Figure 4 shows the concentration level of each features of the dataset after we preprocessed it and filled all the missing values according to the above methodology.

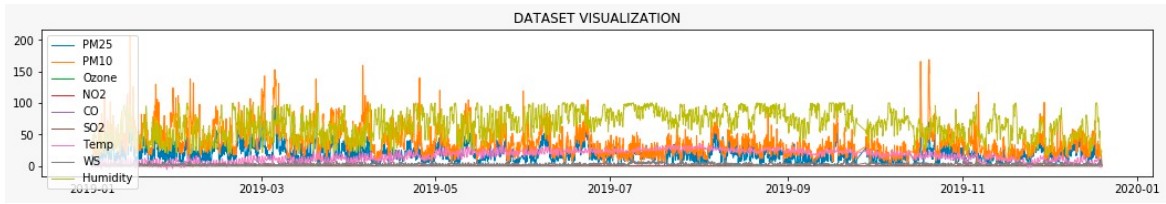

**Figure 4.** Dataset visualization.

As presented in Figure 1, several factors affect the quality of the air, and each factor has its own physical properties and dimensions. A deep analysis of these factors is required in order to achieve a good accuracy and prevent training problems such as overfitting, so in data preprocessing of our framework, we merged both data sources as one dataset, and we normalized that dataset by subtracting the mean of each feature and dividing by the variance of each feature, as presented in the following equation:

$$x_{std}^i = \frac{x^i - x_{mean}^i}{\sigma_x^i}.$$ (5)

In the above equation, $x_{mean}^i$ and $\sigma_x^i$ represent, respectively, the mean and variance of the *i*-th characteristic variable.

### 2.3. Features Correlation

In order to build a proper prediction model for air quality, it is important to understand and identify the correlation between the various factors of air quality index. In our case, since we wanted to predict the concentration level of particles $PM_{2.5}$ and $PM_{10}$, it was very crucial to understand the correlation between each of these two particles with the others features, especially with the meteorological features. Particles $PM_{2.5}$ and $PM_{10}$ are affected by many measurable factors, but not all of them should be used as input for the forecasting task, and the irrelevant factors would become burdensome for the model. In order to see the correlation between each of these particles with the meteorological data, we calculated the correlation coefficient ($\delta$) between each features and the target particle based on the following formula, supposing that we had an observation vector A = $(a_1, a_2, a_3, \ldots, a_n)$ with another vector B = $(b_1, b_2, b_3, \ldots, b_n)$:

$$\delta = \frac{n \sum_{i=1}^n a_i b_i - \sum_{i=1}^n a_i \sum_{i=1}^n b_i}{\sqrt{n \sum_{i=1}^n a_i^2 - \left(\sum_{i=1}^n a_i\right)^2} \sqrt{n \sum_{i=1}^n b_i^2 - \left(\sum_{i=1}^n b_i\right)^2}}.$$ (6)

In the above formula, when $0 < \delta < 1$, there is a positive correlation, and if we have $-1 < \delta < 0$, the correlation will be negative. When the value of $\delta$ is closer to 1, the difference between A and B is smaller and the correlation is higher. In Figure 5 we present the features correlation heatmap of the dataset with the particle $PM_{2.5}$ as target. Besides the feature correlation heatmap, we also created

several pairwise bivariate distribution plots in order to clearly see how each particle is affected by the gas features ($O_3$, $NO_2$, $CO$, and $SO_2$) and the meteorological features (wind speed, temperature, and humidity).

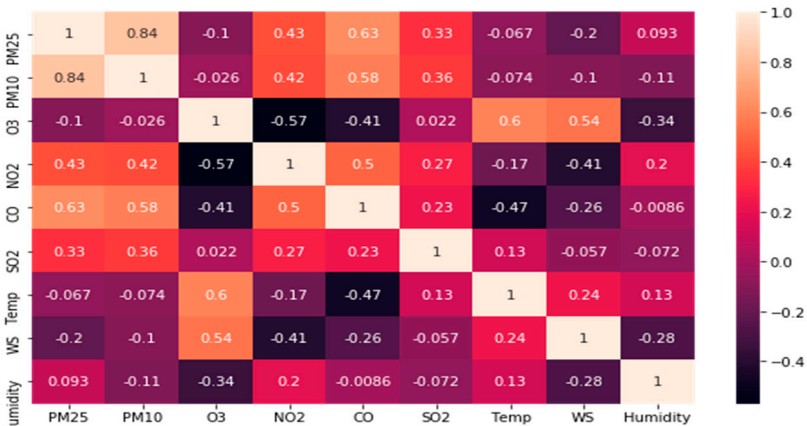

**Figure 5.** Features correlation heatmap.

This analysis was very important because in the proposed approach we propose encoding the key evolution patterns of the meteorological and the gases time series data to provide more information and patterns for the final prediction of both particles.

As depicted in Figures 5 and 6, $PM_{2.5}$ and $PM_{10}$ both had a negative correlation with the temperature and the wind speed, which means that lowering the temperature could increase the concentration level of these particles and result in bad air quality. At the same time, high wind speed will result in normal or good air quality, since the wind will disperse the particles in the air and high humidity usually causes a high concentration of particles. Moreover, high levels of $NO_2$, $CO$, $SO_2$, and a low level of $O_3$ also lead to a high concentration of particles. It was also found that both particles were highly correlated, which indicates that they have very similar patterns, so we did not use both as input to predict one or another in order to avoid redundant information that could lead to overfitting. For example, while predicting the $PM_{2.5}$ we did not use $PM_{10}$ as input, and vice versa. On the other hand, all the meteorological features were weakly correlated with each other, which shows that there is no information replica between them, and they could be directly used as the input of the prediction model.

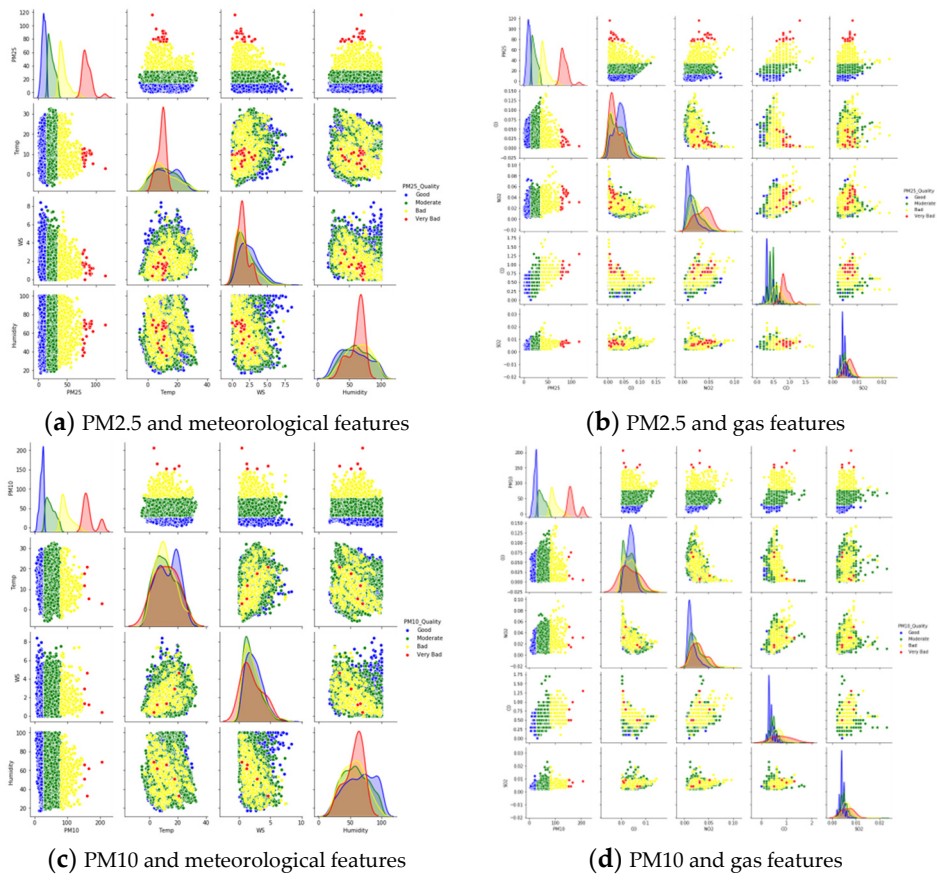

(**a**) PM2.5 and meteorological features

(**b**) PM2.5 and gas features

(**c**) PM10 and meteorological features

(**d**) PM10 and gas features

**Figure 6.** Pairwise distribution analysis.

## 3. Proposed Deep Learning Architecture

Several researchers in literature have studied hybrid deep learning models, which are usually effective for improving the performance of typical deep learning algorithms. In the proposed architecture, we combine a stacked autoencoder, a convolutional neural network, and a bi-directional LSTM layer together to predict the concentration level of $PM_{2.5}$ and $PM_{10}$ based on data collected in Busan metropolitan city. The proposed deep learning framework is based on two stages of deep features extraction. In the first stage we extract suitable features from both particles time-series data, and in the second stage we use a stacked autoencoder layer for encoding the key patterns of meteorological and gas related features. Figure 7 shows the architecture of the proposed model.

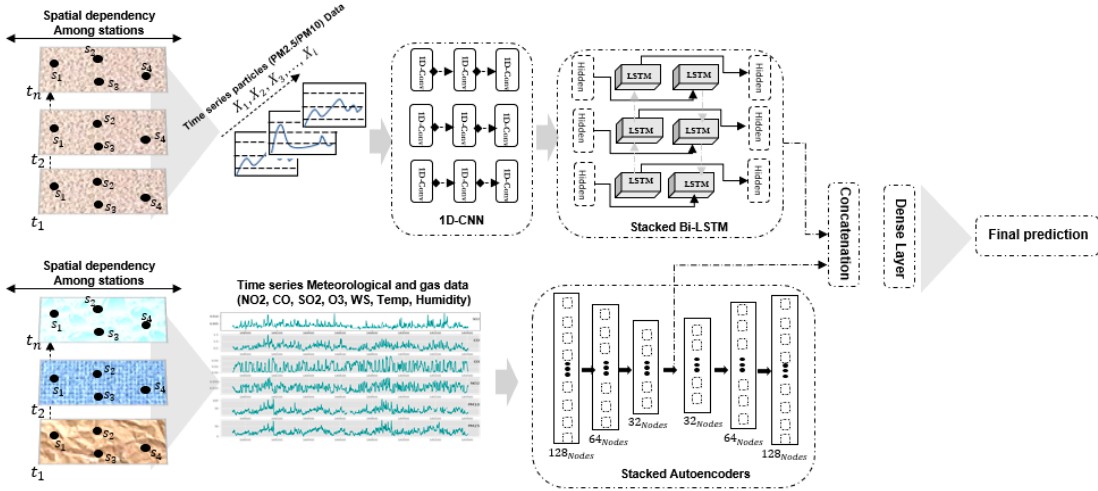

**Figure 7.** Proposed model.

### 3.1. 1D-CNN for Deep Features Extraction on PM$_{2.5}$ and PM$_{10}$ Particles Data

The CNN model is widely used in the object recognition and image processing area, but due to its one-dimensional characteristics it can also be applied to time series forecasting tasks. In our study, the CNN model takes the air quality data in one-dimensional form, wherein the data are shaped in order of sequential time instants. Moreover, considering the properties of multivariate time series air quality data, we also leverage the strength of one-dimensional CNN (1D-CNN) to extract the local pattern features, and the deep spatial correlation features of the 20 air quality monitoring stations present in Busan metropolitan city. A standard CNN model has four layers: input, convolutional, pooling, and output layers. A typical convolution process can be represented by the following equation:

$$X^{(l)} = \phi\left(X^{(l-1)} * W^{(l)} + b^{(l)}\right). \tag{7}$$

$X^{(l)}$ and $X^{(l-1)}$ symbolize the output pattern of the *l*th and (*l* − 1)th layers. $W^{(l)}$ and $b^{(l)}$ are, respectively, the weight and the bias of the $l_{th}$ layer, and $\phi$ is the activation function. To minimize the dimension of data, the CNN network implements a pooling layer after the convolutional layer to improve the model configuration. The pooling layer can select useful data from the input layer. CNN combines convolutional and pooling layers, which is represented through the following equation:

$$X^{(l+1)} = \phi(pool \max X^{(l+1)} * W^{(l+1)} + b^{(l+1)}). \tag{8}$$

In order to represent the spatial-temporal features of **PM$_{2.5}$** and **PM$_{10}$** particles among all the monitoring stations, we pre-trained several one-dimensional CNN layers to select local recurring features and deep spatial similarity features of multiple patterns observed in different stations. Apart from image processing that uses two-dimensional image pixels as CNN's inputs, multiple one-dimensional data are inputted to the first part of our deep learning framework.

### 3.2. Using Stacked Autoencoders for Gas Features and Meteorological Patterns Encoding

There are several important indicators in the air quality forecasting task. Predicting particles such as **PM$_{2.5}$** and **PM$_{10}$** without gas features and meteorological information may result in bad accuracy and therefore bad decision-making, and this is why the large majority of the available dataset related to air quality are not only based on particle matters, but also on meteorological data and gas-related data. Considering that, in this study, we took into consideration the meteorological and the gas features to improve the accuracy of our model while predicting **PM$_{2.5}$** or **PM$_{10}$**. We used a stacked autoencoders network to encode the information from these features. An autoencoder can be viewed as a kind of

neural network typically based on one hidden layer, which aims to set the objective value equal to the input. It usually compresses the input into a latent-space representation, and then reconstructs the output from this representation. Autoencoder networks are unsupervised models including two major processes, which are encoder process and decoder process. These major processes allow this neural network to learn some more abstract features by an unsupervised way. In this paper, we intended to build a vector representation for meteorological and gas information and use it for the final prediction of the $PM_{2.5}$ or $PM_{10}$ particle.

### 3.3. Implementation of Stacked BiLSTM for Capture Temporal Dependencies Considering Forward and Backward LSTM Directions Simultaneously

Traditional methods such as ARIMA and some shallow learning algorithms have suffered poor performance in forecasting tasks due the fact that they do not consider the long-term dependence of time series data. The LSTM network as presented in Figure 8 has been so far the best solution to overcome this shortcoming. In an LSTM model, units can teach the network to learn when to forget historical data and when to update memory units through a new input architecture. The basic structure of an LSTM memory unit is composed of three essential gates, namely, input, forget, and output, as presented in Figure 8. The gates include a sigmoid layer and a pointwise multiplication operation and can control the data flow of LSTM to prevent gradient eruption. The input gate determines the number of new features that will be reserved, the output gate decides the data that will be delivered, and the forget gate regulates the content that will be abandoned from the previous states. The memory units include a historical information form, which is controlled by the three gates. The conventional LSTM block computing process uses the following equations:

$$i_t = \sigma(W_i x_t + U_i h_{t-1} + V_i c_{t-1}), \tag{9}$$

$$f_t = \sigma\left(W_f x_t + U_f h_{t-1} + V_f c_{t-1}\right), \tag{10}$$

$$o_t = \sigma(W_o x_t + U_o h_{t-1} + V_o c_t), \tag{11}$$

$$\widetilde{c_t} = \tan h(W_c x_t + U_c h_{t-1}), \tag{12}$$

$$c_t = f_t{}^\circ c_{t-1} + i_t{}^\circ \widetilde{c_t}, \tag{13}$$

$$h_t = o_t{}^\circ \tan h(c_t). \tag{14}$$

The input gate $i_t$ determines the value that needs to be updated and updates the memory cell $\widetilde{c_t}$. The forget gate $f_t$ identifies the information that is to be forgotten at $t-1$ time with the output value $h_{t-1}$. The output gate $o_t$ and memory cell $c_t$ determine the information that can be output and get the output value $h_t$.

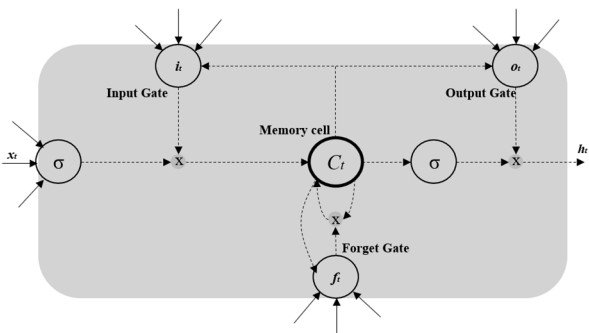

**Figure 8.** LSTM (long short-term memory) network.

One drawback of the traditional is that it only proceeds in a unidirectional way and may cause the loss of significant information when extracting deep suitable features. Therefore, it is important

to utilize both directions of network traffic in order to generate more important features. The goal is to split the state neurons of a standard LSTM into a part that is responsible for the backward states and a forward state. Outputs from forward states are not connected to inputs of backward states, and vice versa. The Bi-LSTM combines the hidden LSTM states of opposite directions to the same output. With this architecture, the output layer will be able to get information from both future and previous states. In this paper, we applied Bi-LSTM, as shown in Figure 9, to capture the temporal dependencies of particle matters between two directions. The Bi-LSTM considers forward and backward LSTMs simultaneously through two independent hidden layers. The outputs of the forward and backward LSTMs are concatenated to compute the output of Bi-LSTM. The hidden states of the forward and backward layers are measured based on the following equations:

$$h_f = o_f{}^\circ \tan h\left(c_f\right),\tag{15}$$

$$h_b = o_b{}^\circ \tan h(c_b).\tag{16}$$

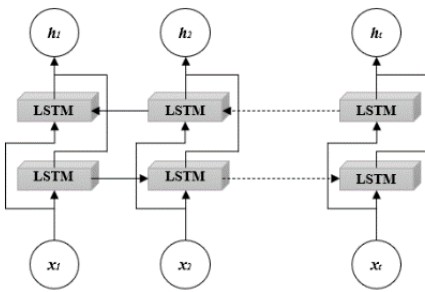

**Figure 9.** Bi-LSTM network.

### 3.4. Using a Distributed Deep Learning Architecture to Train The Proposed Approach

To the best of our knowledge, distributed deep learning training architecture has never been used to challenge the problem of air quality forecasting, especially to reduce the training time and the memory consumption. Our framework was implemented using a distributed deep learning model called data parallelism. We distributed the historical air quality data and the meteorological data across multiple pipelines or nodes before training. The following algorithm represents the training process of the proposed framework (Algorithm 1):

---

**Algorithm 1** Distributed training process

---

1:     **Initialization** of coordinator node parameters
2:     **Define** the number of training nodes N
3:     Dataset portioning into *n* shards
4:     **for** each monitoring station
5:         datashard ← dataset/N
6:     **end for**
7:     **for** each node $i \in \{1,2, 3, \ldots, N\}$ **do**
8:         LocalTrain(Model[parameters], datashard)
9:         $\nabla fi$ ← Backpropagation () //send gradient to coordinator node
10:    **end for**
11:    UpdateMode ← Asynchronous () // Model replicas will be asynchronously aggregated via peer-to-peer communication with the coordinator node
12:    Aggregate from all nodes: $\nabla f \leftarrow 1/N \sum_{i=1}^{N} \nabla fi$
13:     **for** each node $i \in \{1,2, 3, \ldots, N\}$ **do**
14:         CoordinatorNode.Push (ParametersUpdate)
15:    **end for**

---

In order to reduce the training time and produces a significant computing performance we used a distributed training process called data parallelism by having *n* training workers optimize a central proposed model by processing *n* different shards (partitions) of the dataset in parallel. In this setting, we distributed *n* model replicas over *n* processing training nodes. Therefore, every node held one model replica. Then, the workers trained their local replica using the assigned data shard. A parameter server was responsible for the aggregation of model updates, and parameter requests coming from different workers. The final flow of the proposed approach is presented in Figure 10.

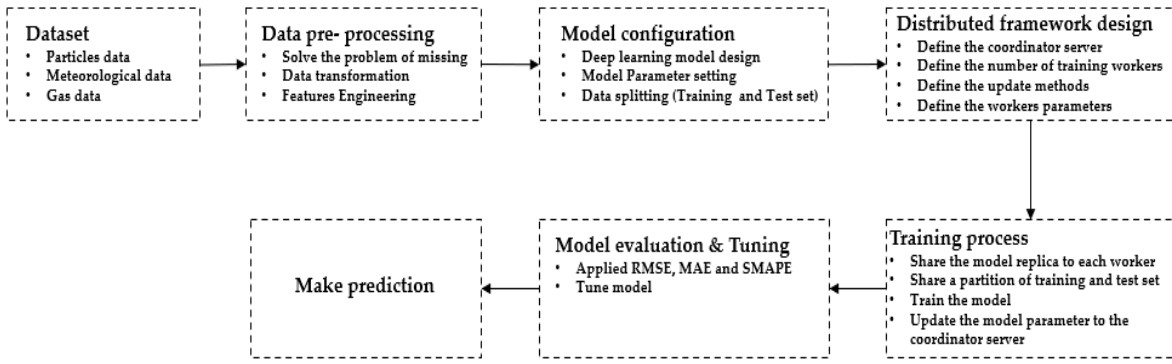

**Figure 10.** Flow of proposed approach.

## 4. Experiment Evaluation

### 4.1. Experiments Setup

This section describes the hardware and software environment of the experiments. We conducted our experiments on a PC Server, with an AMD Ryzen 7 2700x 8-core processor 3.7 GHz processor, two GPUs (NVIDIA GeForce RTX 2080), and 32 GB memory. The proposed framework was implemented using python programming language, we used TensorFlow which is an open source deep learning library to build our learning model, and the other deep learning state-of-the-art models. The library PySpark from the software apache spark was used to perform a distributed computing platform. Scikit-learn was adopted to build shallow learning models. Evaluation of performance was conducted using sklearn metrics. The optimization method was Adam optimizer. In all experiments, the training data and the testing data accounted for 80% and 20% of the dataset, respectively. We compared our framework against 10 state-of-the-art models, which were:

- ARIMA [2]: autoregressive integrated moving average (ARIMA) is a time series prediction model which combines moving average and autoregression components;
- Random Forest regression [8]: a machine learning algorithm which aims to generate every tree in the ensemble from a sample with replacement (bootstrapping) from the training set;
- GBDT [8]: Gradient Boosting Decision Tree (GBDT) is a powerful and widely used machine learning method in time series data;
- Lasso: Lasso is a popular regression analysis algorithm that performs both variable selection and regularization;
- SVR: support vector regression is a machine learning method used for time series forecasting. This method is based on five specific kernels, which are linear, poly, RBF, sigmoid, and precomputed. In this experiment we used the linear kernel to predict the air quality and compare with the proposed approach;
- ANN multi-layer perceptron regression: a multilayer perceptron (MLP) is a class of feedforward ANN that learns a function $f(.) : R^m \rightarrow R^0$ by training on a dataset, where *m* is the number of dimensions for input and 0 is the number of dimensions for output;

- RNN: the recurrent neural network is a specific kind of neural network that allow previous output to be used as inputs while having hidden states;
- LSTM [11]: long short-term memory network (LSTM) is a special kind of recurrent neural network, widely used for time series forecasting;
- Stacked deep autoencoder (DEA) [15]: autoencoder is a kind of unsupervised learning structure that owns three layers: input layer, hidden layer, and output layer;
- CNN+LSTM [12]: hybrid model based on convolution neural network and LSTM.

We evaluated the performance of our model using the following evaluation metrics:

- MAE (mean absolute error): mean absolute error is used to measure the average magnitude of the errors in a set of data values (predictions), without any consideration of direction. This metric is based on the following statement:

$$MAE = \frac{1}{n} \sum\nolimits_{i=1}^{n} |y_i - \hat{y}_i| \tag{17}$$

- RMSE (root mean square error): the root mean square error is used to aggregate the magnitudes of the errors in predictions for various times into a single measure of predictive power. It is a measure of accuracy, which is widely used to compare the prediction error of multiple models for a specific dataset:

$$RMSE = \sqrt{\frac{1}{n} \sum\nolimits_{i=1}^{n} (y_i - \hat{y}_i)^2} \tag{18}$$

- SMAPE (symmetric mean absolute percentage error): the SMAPE is an accuracy metric based on percentage errors. This metric can be defined as follows:

$$SMAPE = \frac{1}{2n} \sum\nolimits_{i=1}^{n} \frac{|y_i - \hat{y}_i|}{(|y_i| + |\hat{y}_i|)} \times 100\%. \tag{19}$$

We divided our experiments into two phases. For the first phase we trained our model on a centralized deep learning architecture, as usually happens in the literature, and compared it with the-state-of-the-art methods that we mentioned above. For the second phase the proposed deep learning framework was trained based on a distributed deep learning architecture called data parallelism. In this phase we used 10 workers to train 10 different shards of the meteorological, gas, and air quality data in parallel. Each worker held a replica of the proposed framework. We further used a parameter node for the aggregation of model updates, and parameter requests coming from different workers.

*4.2. Experiment Setup*

4.2.1. Phase 1

In phase 1, we predicted the concentration level of the particle $PM_{2.5}$ and $PM_{10}$ based on the historical air quality and meteorological data of Yeonsan-dong station. We used three layers of encoders with the sigmoid function as activation function and three layers of decoders with Relu function. The stacked autoencoders and decoders had different hidden layers, they were based on {128, 64, 32} nodes for the encoder parts and {32, 64, 128} nodes for the decoders. After pre-training SAE on meteorological and gas data, we could obtain rough compressed data from final hidden layer of corresponding SAE. These data were consequently concatenated with the output layer of the Bi-LSTM network to make the final prediction. Table 1 summarizes the comparison results of different algorithms. As shown in the table, the proposed framework was superior to other baseline models in terms of $PM_{2.5}$ and $PM_{10}$ prediction performance. The accuracy of the GBDT model was the lowest for both particles' prediction. It had the highest MAE, RMSE, and SMAPE values, which were, respectively,

16.17, 22.96, and 33.59 for $PM_{2.5}$ prediction, and 23.05, 25.55, and 34.17 for $PM_{10}$ prediction. The MLP, SVR, RFR, and the LASSO algorithms also had high error rates. These five models performed poorly while predicting both particles $PM_{2.5}$ and $PM_{10}$. The ARIMA performed a little bit better compared to the previous one. It got 9.03, 13.49, and 27.81, respectively, for MAE, RMSE, and SMAPE evaluation metrics while predicting $PM_{2.5}$. On the other hand, 9.34, 12.89, and 27.78, respectively, for MAE, RMSE, and SMAPE evaluation metrics while predicting $PM_{10}$. The classic neural networks algorithms like RNN, DAE, and LSTM were between 7.19 and 7.90 for the MAE, 8.93 and 10.02 for RMSE, and 23.15 and 24.17 for SMAPE metric for the $PM_{2.5}$ forecasting. For $PM_{10}$ forecasting the results were similar. Finally, the hybrid model based on the convolutional neural network and LSTM had the best performance among the state-of-the-art models in both predictions.

**Table 1.** Models' evaluation results.

| Models | $PM_{2.5}$ | | | $PM_{10}$ | | |
|---|---|---|---|---|---|---|
| | **MAE** | **RMSE** | **SMAPE** | **MAE** | **RMSE** | **SMAPE** |
| GBDT | 16.17 | 22.96 | 33.59 | 23.05 | 25.55 | 34.17 |
| MLP | 15.63 | 19.03 | 32.11 | 22.38 | 19.91 | 36.09 |
| SVR | 14.28 | 18.72 | 33.17 | 15.71 | 19.32 | 34.18 |
| RFR | 12.73 | 16.31 | 29.41 | 13.64 | 14.63 | 30.41 |
| LASSO | 10.35 | 13.99 | 28.89 | 12.31 | 14.21 | 29.37 |
| ARIMA | 9.03 | 13.49 | 27.81 | 9.34 | 12.89 | 27.78 |
| RNN | 7.90 | 10.02 | 24.17 | 8.47 | 12.33 | 25.23 |
| DAE | 7.89 | 10.31 | 24.29 | 8.10 | 11.17 | 24.94 |
| LSTM | 7.19 | 8.93 | 23.15 | 6.99 | 9.33 | 24.61 |
| CNN + LSTM | 5.90 | 8.33 | 21.03 | 6.21 | 8.27 | 21.93 |
| **Proposed Approach** | **5.07** | **6.93** | **18.27** | **5.83** | **7.22** | **17.27** |

However, our approach outperformed even the hybrid model for both particles' predictions. As we can see, the deep learning models performed better than the classic machine learning algorithms in general. By this, we can conclude that deep learning-based approaches are more efficient than machine learning approaches for air quality forecasting task, and the hybrid models are more efficient than the standard deep learning models.

In addition, we investigated the impact of each epochs on different deep learning models, as presented in Figure 11. It can be seen how our method maintained the best performance compared to the other models, especially while predicting $PM_{2.5}$ particles. During the prediction of $PM_{10}$ the proposed model had a lesser error rate for the first 10 epochs, from epoch 17 to 24, the hybrid CNN + LSTM had the lesser error rate, and at the end we can see how the MAE value of the proposed model decreased again and was lesser than the other algorithms. Furthermore, in this experiment, for more detailed comparison, we plotted the prediction performance of the proposed model against the three best performing baseline models (DAE, LSTM, and CNN + LSTM) while predicting both particles $PM_{2.5}$ and $PM_{10}$. As represented in Figures 12 and 13, in contrast to the baseline models, the prediction pattern of the proposed model was very similar to the real pattern of the time-series data for both particles.

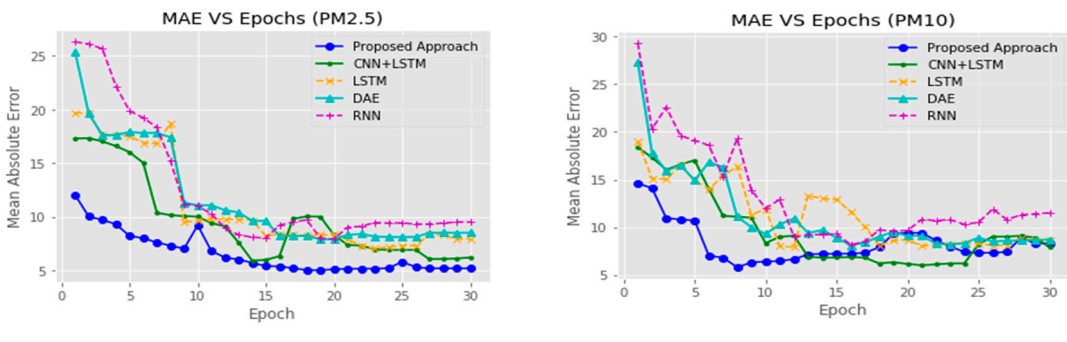

**Figure 11.** MAE (mean absolute error) of the proposed approach vs different epochs and comparison with others deep learning models.

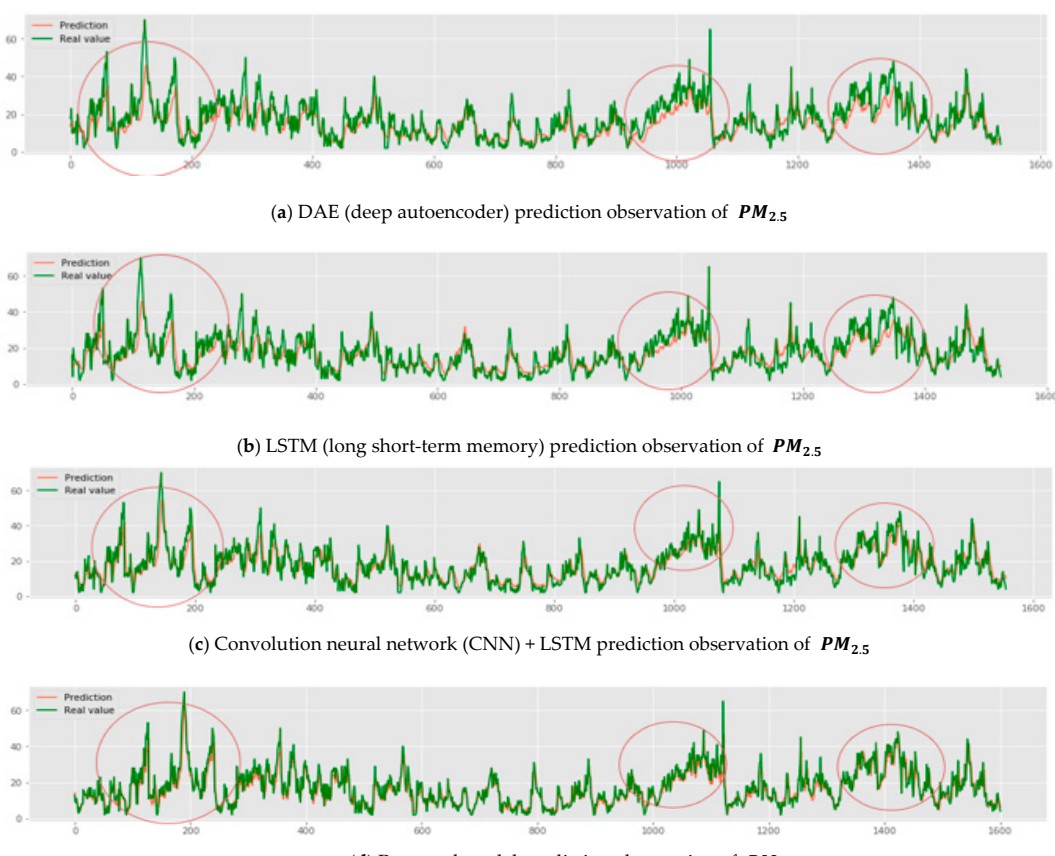

(**a**) DAE (deep autoencoder) prediction observation of $PM_{2.5}$

(**b**) LSTM (long short-term memory) prediction observation of $PM_{2.5}$

(**c**) Convolution neural network (CNN) + LSTM prediction observation of $PM_{2.5}$

(**d**) Proposed model prediction observation of $PM_{2.5}$

**Figure 12.** Prediction observation of $PM_{2.5}$ for the proposed approach and comparison with others deep learning models.

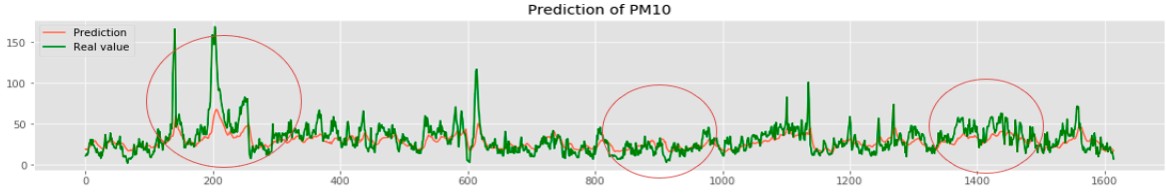

(**a**) DAE prediction observation of $PM_{10}$

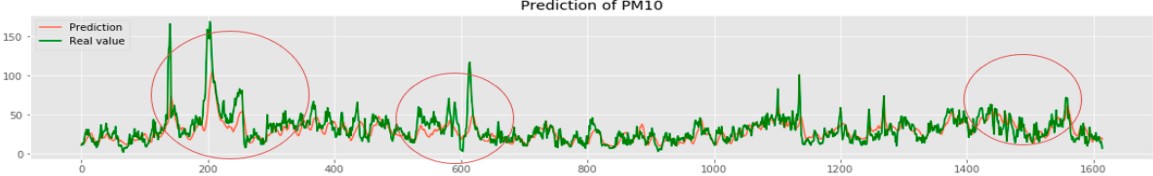

(**b**) LSTM prediction observation of $PM_{10}$

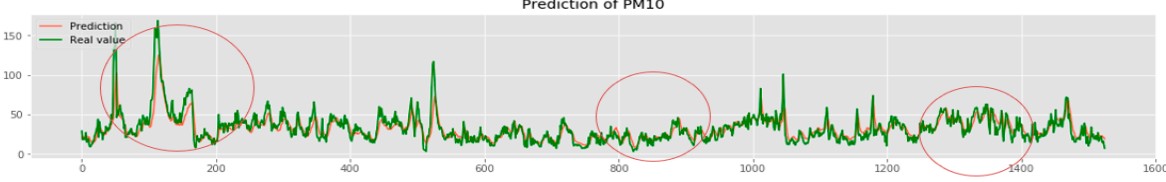

(**c**) CNN + LSTM prediction observation of $PM_{10}$

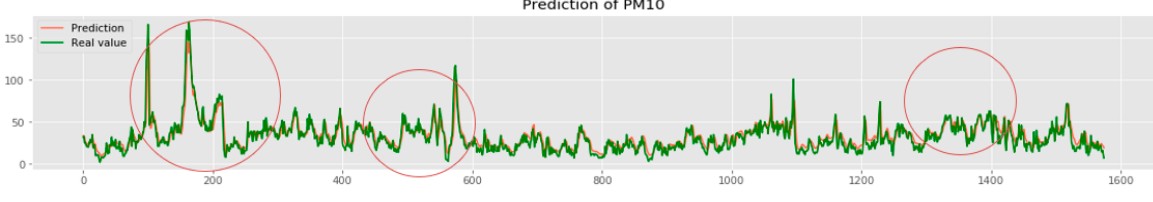

(**d**) Proposed model prediction observation of $PM_{10}$

**Figure 13.** Prediction observation of $PM_{10}$ for the proposed approach and comparison with others deep learning models.

The prediction curve of the proposed model increased quickly, so it could forecast the particles more accurately; on the other hand, baseline models, especially the deep autoencoders model, failed to track the trends of $PM_{2.5}$ and $PM_{10}$. In this experiment the difference between both patterns (prediction and real) can be seen more clearly during the time period of wave peak (see Table 2).

**Table 2.** Training time evaluation.

| Models | Average Training Time (s) |
|---|---|
| RNN | 482 |
| DAE | 463 |
| LSTM | 345 |
| CNN + LSTM | 341 |
| **Proposed model** | **310** |

Then, we found in our experiment that the proposed framework had not only the best accuracy compared to the baseline models but it also had the best training average times, as it averaged 310/s while making the prediction $PM_{2.5}$. The RNN model recorded the highest average training time, and

the LSTM and the hybrid model had almost the same average time. In the following we will optimize the performance of the proposed model, especially its training, by using a distributed architecture.

4.2.2. Phase 2

In this phase, we trained our model based on Algorithm 1. We used ten specific training workers to perform this task. The replica of the proposed was dispatched over each worker with the model's parameters. We intended to predict the same particles using the same data as phase 1. As shown in Figure 14, the proposed model performed the best for both particles' prediction while training it using six workers.

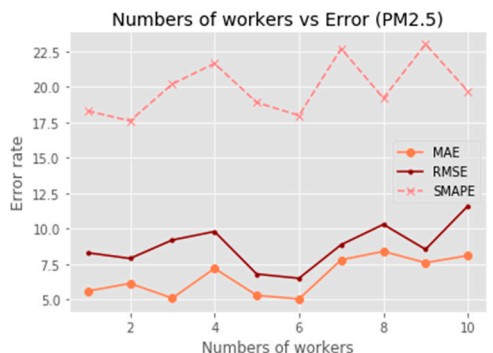 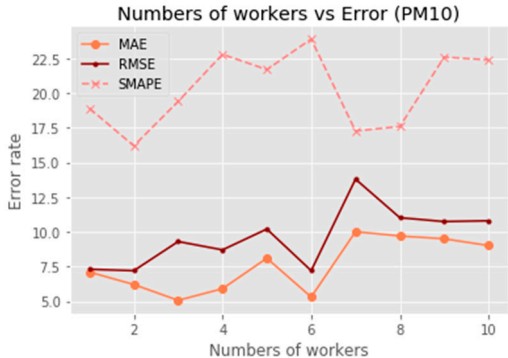

**Figure 14.** Evaluation of the proposed model while using the distributed training process.

For instance, during the prediction of $PM_{2.5}$, when the number of training workers reached six, the MAE value was 5, the RMSE was 7.1, and the SMAPE 17.5, each of these values were recorded as the lowest among all training workers' numbers. These error values were even similar to the ones recorded in phase 1. Also, we could observe a similar result happening for the predictions of particle $PM_{10}$, which recorded the values 5, 7.5, 22.8, respectively, for MAE, RMSE, and SMAPE. By looking at this experience, we can conclude that the right number of nodes for the prediction of these two air quality particles is six. Furthermore, we were also interested in analyzing the training time of the proposed model during this second experimental phase.

As depicted in Figure 15, the training time of our model decreased while the number of workers increased. At four workers we reached 200 s, which was the lowest training time. At six workers the time increased a little bit to 220 s and then decreased to 210 s and stayed constant. Compared to the centralized training in phase 1, the training time in phase 2 also decreased two times, which makes our methods more effective.

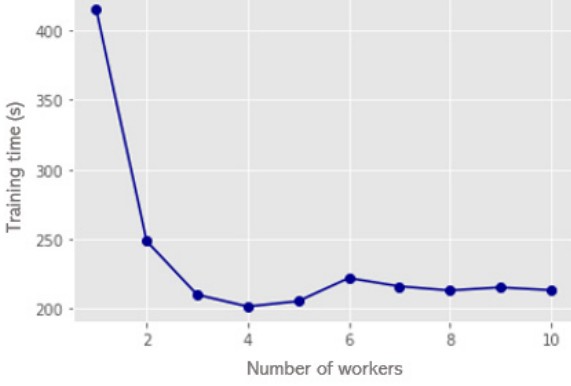

**Figure 15.** Training time evaluation.

## 5. Conclusions

In this paper we proposed a convolutional Bi-LSTM autoencoder model for air quality prediction in Busan metropolitan city. The proposed approach utilizes the historical $PM_{2.5}$ and $PM_{10}$ time series data with the encoded meteorological and gas pollution data to perform the prediction. We used a two-stage feature extraction method to allow our model to perform well in any atmospheric condition (stable or unstable). In the first stage a stacked autoencoder extracts useful features and information from pollution gas data and meteorological data, and in the second stage a convolutional layer is used to extract deep spatial correlation features from air quality particles data. The training process of the proposed approach was based on a distributed learning method, namely data parallelism, in which several training workers were leveraged for training each partition of the air quality dataset by using a model replica. A coordinator node was responsible for the aggregation of model updates, and parameter requests coming from different workers. The experiments were conducted based on one-year air quality data collected in Busan city. The training data and the testing data accounted for 80% and 20% of the dataset, respectively, which means almost two months for testing data. The experimental results showed the superiority of the proposed model over ten state-of-the-art models. The proposed model recorded the lowest error rate while predicting both particles. For $PM_{2.5}$ prediction, we registered 5.07, 6.93, and 18.27, respectively, for MAE, RMSE, and SMAPE evaluation metrics, and for $PM_{10}$ forecasting we recorded 5.83, 7.22, and 17.27, respectively, for the same metric. Also, it was found that the distributed training method can significantly improve the training time, which can help the government to implement appropriate emergency measures to mitigate atmospheric pollution in a short period of time.

**Author Contributions:** A.G.M.M. investigated the theoretical basis for this work, implemented the machine learning and the deep learning models for the experiments as well as the proposed approach, and wrote the manuscript. Y.K. collected the data used in the experiments section and made some adjustment in the data pre-processing part. Y.Y. supervised the entire work and helped to write the manuscript, and revised the manuscript. J.A. revised the manuscript, supervised the research and administrated the project. All authors have read and agreed to the published version of the manuscript.

**Funding:** This research was supported by a grant from the Technology Advancement Research Program (grant No. 20CTAP-C152124-02) funded by the Ministry of Land, Infrastructure, and Transport of the Korean government.

**Conflicts of Interest:** The authors declare no conflict of interest.

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
