# Peer review of "Distributed Deep Features Extraction Model for Air Quality Forecasting"

_sustainability, doi:10.3390/su12198014_

Round 1

Reviewer 1 Report

The authors use deep learning methods to predict air quality (e.g., the concentration of the air quality particles) based on time series data. The work is significant and interesting. 

The authors use CNN and Bi-LSTM layer to learn the spaito-temporal correlation of air quality. LSTM is for time series /sequence data. However, with my limited knowledge, CNN in general is used for image study. How CNN was applied to the current study? What's the technics behind this?

I am confused. The new roposed method is based on CNN and Bi-LSTM layer (stated in the abstract), while the statements on Page 3 and 4 in the section of Introduction metion that the method is based on Bi-LSTM. 

It may be better to give a name of the proposed approach. 

Reviewer 2 Report

The article focuses to the  air quality forecasting. The interesting distributed deep features extraction model  is proposed. The problem is not new, but there are no single solutions yet. This direction is developing intensively in the world.

A significant note in the article is the section of conclusion. In our opinion, the discussion presented in conclusion is not enough. Conclusions should be formed more clearly. In the conclusion section, the characteristics of the proposed model should be given in numerical form (for example, authors must provide errors). In addition, it would be useful to report if the model has limitations in conditions of unstable or stable stratification of the atmosphere. Time scales should also be indicated in the conclusions (for example, the time averaging of used data, the maximum forecast period).

In general, the article is interesting and can be published after correcting the conclusions.

Round 2

Reviewer 1 Report

The explanation of CNN applied to time series data is good for readers to understand.

However, the writing at the end of page 3 and beginning of page 4 is still difficulty to understand and not well-organized. 
